# Estimation of ANT-DBS Electrodes on Target Positioning Based on a New Percept^TM^ PC LFP Signal Analysis

**DOI:** 10.3390/s22176601

**Published:** 2022-09-01

**Authors:** Elodie Múrias Lopes, Ricardo Rego, Manuel Rito, Clara Chamadoira, Duarte Dias, João Paulo Silva Cunha

**Affiliations:** 1INESC TEC—Instituto de Engenharia de Sistemas e Computadores, Tecnologia e Ciência, Faculty of Engineering, University of Porto, 4200-465 Porto, Portugal; 2Neurophysiology Unit, Neurology Department, Centro Hospitalar Universitário de São João, 4200-319 Porto, Portugal; 3Neurosurgery Department, Centro Hospitalar Universitário de São João, 4200-319 Porto, Portugal

**Keywords:** ANT-DBS, LFPs, target localization, epilepsy, closed-loop stimulation

## Abstract

Deep brain stimulation of the Anterior Nucleus of the Thalamus (ANT-DBS) is an effective therapy in epilepsy. Poorer surgical outcomes are related to deviations of the lead from the ANT-target. The target identification relies on the visualization of anatomical structures by medical imaging, which presents some disadvantages. This study aims to research whether ANT-LFPs recorded with the Percept^TM^ PC neurostimulator can be an asset in the identification of the DBS-target. For this purpose, 17 features were extracted from LFPs recorded from a single patient, who stayed at an Epilepsy Monitoring Unit for a 5-day period. Features were then integrated into two machine learning (ML)-based methodologies, according to different LFP bipolar montages: Pass1 (nonadjacent channels) and Pass2 (adjacent channels). We obtained an accuracy of 76.6% for the Pass1-classifier and 83.33% for the Pass2-classifier in distinguishing locations completely inserted in the target and completely outside. Then, both classifiers were used to predict the target percentage of all combinations, and we found that contacts 3 (left hemisphere) and 2 and 3 (right hemisphere) presented higher signatures of the ANT-target, which agreed with the medical images. This result opens a new window of opportunity for the use of LFPs in the guidance of DBS target identification.

## 1. Introduction

Epilepsy is the second most common neurological disease, affecting around 0.5% of the population worldwide [1]. Treatment with adequately chosen antiseizure drugs allows for adequate control in up to 60% of patients, but the remainder have persisting and frequently disabling seizures (drug-resistant epilepsy, DRE) [2]. For a minority of those patients, resective surgery is a safe and effective option, but a substantial number of patients do not achieve long-lasting remission [3]. Moreover, many patients with DRE are not suitable candidates for cortical resections due to overlapping eloquent cortex or multifocal seizure onsets. Neurostimulation is often considered in these patients, including vagal nerve stimulation or deep brain stimulation (DBS). For adults with focal DRE, the DBS of the anterior nucleus of the thalamus (ANT) is an established approved treatment [4].

The ANT has long been considered a suitable DBS anatomical target for patients with refractory epilepsy, due to its therapeutic action on the interruption of seizure spread [5,6,7,8]. The ANT is in the superior region of the thalamus and is separated from the rest of the thalamus by the anterior medullary lamina. It consists of three subnuclei designated as anteroventral (AV), anterodorsal (AD) and anteromedial (AM) nuclei [9]. After the first ANT-DBS surgery performed by Cooper and Upton in 1980 [10,11,12], several studies have reported the therapeutic efficacy as well as the safety of this procedure [13,14,15,16,17,18]. The larger-scale multicenter trial, the SANTE trial, has reported a seizure reduction of 56–69% [19,20].

It is generally accepted that stimulation of the ANT is more effective compared to stimulation outside this region, since lead locations deviating from the target region (the AV nucleus) have been related with poorer surgical outcomes [21,22]. Furthermore, it is known that changes in stimulation parameters (current voltage, pulse width, frequency and stimulation cycling) produce a minimal effect compared with the choice of the active contact selection [21,22]. This highlights the importance of an accurate identification of the precise location of stimulation.

The identification of the DBS target relies on the visualization of anatomical structures in magnetic resonance imaging (MRI), which can be performed indirectly or directly [21]. In the indirect methodology, the target is defined in the brain atlas using common landmarks, such as the anterior and posterior commissures; the direct method can be performed using 3T MRI techniques [23], which allow for the direct visualization of white-matter structures involving the ANT, such as the external medullary lamina (EML), the internal medullary lamina (IML) and the mammillothalamic tract (MMT) [23,24]. However, both methods present some disadvantages: the indirect method is limited due to anatomical variations of the target structure in the stereotactic space between individuals [23,24]; the direct method, on the other hand, requires advanced imaging techniques, which may be limited in some DBS centers [23].

Several studies have investigated the potential of complementary methods in the guidance of the target identification, such as the analysis of single-unite (SU) recordings during DBS [21,24,25,26,27]. Hodaie et al. were the first to describe the electrophysiological properties of SU signals from the ANT of anesthetized patients during DBS along transventricular trajectory. They found bursting activity characterized as low-threshold calcium spikes (LTS), which were mostly observed during sleep [25]. Later, Schaper et al., found the same pattern using an extraventricular trajectory to the ANT [27]. The same result was also reported by Mottonen et al. [28]. Considering both trajectories, the same authors showed that SU signals were able to distinguish between the ANT from the ventral anterior nucleus [28]. In all these studies, an increase in the firing rate was found at the entrance of the ANT, and a decrease was found when exiting this structure. Despite these advances, it is still unknown whether there is a relation between the neuronal firing properties of the ANT and the clinical DBS outcome in epilepsy [27,28]. On the other hand, LTS bursts were not specific to ANT, since they were also found in other structures, such as the circularis and dorsomedial nucleus of the thalamus [27]. This highlights the need to find new methods that can complement the existing techniques for DBS targeting.

Local field potentials (LFPs) can be recorded by DBS leads and reflect the synchronous pre- and postsynaptic activity of neural populations [29,30,31,32,33,34]. Unlike SU recordings, LFPs can detect focal network rhythms and can be recorded both intra and extraoperatively [35]. Temporary recordings of LFPs can be performed by externalized DBS leads (e.g., [36]), but this recording type presents disadvantages, such as the tome restriction of recording, the influence of microlesions provoked by the oedema around the lead [37] and the increased risk of infection [38]. Recently, the Medtronic company (Medtronic Inc. (Dublin, Ireland)) launched a new neurostimulator device, the Percept^TM^ PC, which can record LFPs at the same time it stimulates, whether the patient is in the hospital or not (UC202013078EE©Medtronic2020). Compared to the previous systems (e.g., the Active TM PC+S), the Percept^TM^ PC system has longer battery life and is also able to stream data continuously in real time, as well as to correlate data with patient logged events.

To understand the potential role of DBS-LFPs for DBS targeting, a literature survey in PubMed and Web of Science was conducted, following the keywords (“Percept^TM^ PC” [tiab] or “Activa^TM^ PC+S” [tiab] and “Local Field Potentials”). A total of 12 human-based studies were found [39,40,41,42,43,44,45,46,47,48,49,50]. These studies are summarized in Table 1. Most of them research and report biomarkers of motor fluctuations in Parkinson’s disease (PD). These biomarkers can be found by extracting features from LFPs. The most common type of features can be categorized into spectral (features extracted in the frequency domain reflecting potential fluctuations associated with neural activity); morphological (features describing the signal morphology); statistical (features describing signal variance distribution) and multivariate (features that capture correlations between channels) [51]. For PD, LFP power density was the most frequent feature extracted since beta activity has been seen to correlate with PD symptoms. Regarding epilepsy, no study was found. Moreover, none of these studies focus of the study of LFPs in the optimization of DBS targeting.

In this work, we aimed to study whether ANT-LFPs recorded extraoperatively can be an asset in the identification of DBS target structures, to complement structural imaging approaches, making its joint usage a swifter and more reliable procedure. For this purpose, LFP signals were recorded from a single epilepsy patient, who stayed at an epilepsy monitoring unit (EMU) for 5 days, for simultaneous video-electroencephalography (vEEG) and Percept PC-LFP recordings. From time-domain LFP signals recorded during periods with no stimulation, 17 features were extracted and then integrated into two machine learning (ML)-based methodologies, developed to identify which led contacts (in both hemispheres) show a “signal-signature” of the ANT target. This paper aims to present the processing and classification methodology and discuss the results and their capability to support neurosurgeons to use LFPs an surrogate marker of the AV nucleus position. To the best of our knowledge, we present the first-ever study to use multichannel LFP signals, collected months after electrodes’ implantation (already out of the influence of the microlesion effect), to identify the best-targeted contacts of the implanted electrodes.

## 2. Materials and Methods

### 2.1. Subject

A 54-year-old right-handed patient, diagnosed with focal refractory epilepsy of hypoxic-ischemic etiology with bilateral perisylvian ulegryc lesions, was admitted to the epilepsy monitoring unit (EMU) of the University Hospital S. João, Porto, Portugal, for simultaneous vEEG and ANT-DBS recordings, over 5 days (20–24 July 2020). This monitoring took place approximately one month after a bilateral ANT-DBS implantation. The patient’s seizures were markedly active by sleep and were characterized by bilateral spasms or tonic posturing, sometimes follows by hyperkinetic movements, and left or bilateral clonic activity. Ictal patterns in scalp EEG were in the midline frontocentral regions or obscured by muscle artefacts; interictally, rare epileptiform discharges were seen over the same region. Background activities awake and asleep were normal. This study complies with the requirements of medical confidentially according to the Declaration of Helsinki and data protection. The patient’s data were pseudonymized to not reveal names, initials, birthdays or other private information.

### 2.2. DBS Lead Contact Localization

The DBS leads were automatically detected by neurosurgeons with the “Lead localization” module of a commercially available software (Elements, BrainLab, Munich, Germany), using preoperative MRIs (fast gray matter acquisition T1 inversion recovery (FGATIR) 3T MRI and fluid attenuated inversion recovery (FLAIR) T2 MRI) and postoperative computed tomography (CT) (Figure 1). Each DBS lead included four contacts (0, 1, 2 and 3) and was implanted via transventricular trajectory. The ANT target of each hemisphere was directly identified by the visualization of the MMT. By inspecting the image, the most inserted contact in the left target is the 3L; and in the right, the 2R.

### 2.3. LFP Recording

Time-domain LFPs were recorded using the BrainSense Survey mode of the Percept^TM^ PC system, which is used to give a broad spatial overview of LFP signals during stimulation from which clinicians usually select a frequency band of interest for each patient for chronic recordings (UC202013078EE©Medtronic2020). The Survey mode acquired data in 6 electrode channels (sampling frequency of 250 Hz) in bipolar reference, according to two types of combinations, i.e., Passes: Pass1 (channels 0–3, 1–3, 0–2), which are stimulation-compatible pairs; and Pass2 (channels 0–1, 1–2, 2–3), which are immediately adjacent pairs. A total of 11 BrainSense Surveys were performed during the patient’s EMU stay, which corresponds to a total of 231 s of time-domain LFPs, recorded in each Pass. Pass1 and Pass2 data were then divided in segments of 5 s, making a total of 46 segments for each Pass.

### 2.4. Dataset Selection

In this work, we aimed to assess whether LFP signals are capable of distinguishing between anatomical structures underlying lead contacts from which LFPs were recorded. For this purpose, we analyzed and compared LFPs recorded from contacts that were mostly inserted into the ANT target and contacts farthest from this target, using an ML-based methodology. Considering Figure 1, contacts 3L and 3R were closer to the target and contacts 0L and 0R were farthest. Since LFP signals were recorded using a bipolar reference, signal spatial locations had to be considered as the midpoints between pairs of contacts [52], to assign each recorded signal to the respective anatomical location. From Pass1 and Pass2 reference combinations, it was possible to identify 5 LFP spatial locations at each hemisphere, indicated as A, B, C, D and E (Figure 2B). The corresponding channels are summarized in Table 2, as well as the proportion of the ANT target underlying each contact and a qualitative measure created to express the level of involvement of each location in the target, based on medical imaging analysis.

This patient was implanted with the Medtronic electrode model 3389, which contains contacts with a length of 1.5 mm separated from each other by 0.5 mm (Figure 2A). Depending on the type of bipolar reference used, the distance between each contact pair to its spatial location (d) varies for Pass2, d = 1 mm for all possible combinations; and for Pass1, d = 2 mm for 0–2 and 1–3 pairs, or d = 3 mm for the 0–3 pair. Due to these differences, we considered the Pass1 and Pass2 signals to be of a different nature. Therefore, we developed two ML-based models for differentiating on-target (signals with physical locations completely inserted in the target: +++) and off-target signals (signals with physical locations completely outside the target: −). The general methodology used is in Figure 3.

For the Pass1 model, we assigned LFP signals with spatial locations D_Right_ as on-target and with B_Left_ as off-target; for the Pass2 model, LFPs with spatial locations E_Right_ were assigned as on-target and with A_Left_ as off-target. For each model, corresponding 5 s-segmented signals were used for feature extraction.

### 2.5. Feature Extraction

A total of 17 features from different feature types were extracted (Table 3). These features were selected according to their consensual significance in the epilepsy state of the art [51,53]. All features were extracted using MATLAB R2016b. Statistical features were extracted using the EEG Feature Extraction Toolbox, while the remainder were extracted using MATLAB custom-made functions. As multivariate features, the node strength (the sum of weights links connected to the node) was computed of each adjacency matrix estimated with metrics present in Table 3. Node strength values were computed using the MATLAB Brain Connectivity Toolbox.

All features were ranked for each dataset, using the MATLAB rankfeatures function, which used an independent evaluation criterion for binary classification (*t*-test pooled variation estimation), to assess which features were more discriminative for the two considered classes (Figure 4). Since there was no agreement on the most relevant features for each dataset (set of features extracted from each in- and off-target spatial location, according to each Pass), a dimensionality reduction procedure through a projection method was performed.

### 2.6. Feature Reduction

While a high number of features may help to discriminate classes, there is a greater potential for overfitting, especially in small datasets. Moreover, irrelevant features can also blur the boundaries between classes [54]. For dimensionality reduction, principal component analysis (PCA) was used, a feature projection method that uses orthogonal transformations for transforming a set of observations of possibly correlated variables into a set of linearly uncorrelated variables (principal components) [7]. The PCA was performed using the pca MATLAB function. The minimum number of principal components (PC) in each Pass-based methodology was chosen as the minimum number that kept 99% of the variance of the entire dataset [55]. The dataset transformed in the reduced space was then used for classification.

### 2.7. Classification

The classification was performed according to two methods: Pass1 method (Pass1 LFP signals) and Pass2 method (Pass2 LFP signals). For each dataset in the reduced space, we assigned each sample with a class label 0 (off-target) and 1 (in-target). Then, each dataset was split into 80% for training and 20% for test. The training set was also split into 80% for a second set of training (training set 2) and 20% for cross-validation (CV dataset). In training set 2, three classification models were trained using standard parameters (Table 4) and tested in the CV dataset. The best classification model (with higher accuracy (acc)) was then retrained in the training set 2 dataset and tested in the CV dataset with varying model parameters, to identify the best parameter (the parameter for which the classifier had higher accuracy).

The Pass1 dataset was reduced into 13 principal components and neural networks with a layer size of 2 were considered the best model and parameter to discriminate classes (acc = 64.29% and acc = 100%, respectively). The Pass2 dataset was reduced into 11 principal components and the kNN with a squared Euclidean distance (seuclidean) were considered the best model and classification model (acc = 94.44% for bot classifications). These models were used to train and test the training and the test sets of each dataset. To assess and compare the classification performance of each method in the discrimination of signals coming from in- and off-target physical locations, we computed the accuracy (number of corrected classifications divided by the total number of classifications), sensitivity (describes the effectiveness of the classifier to classify correctly in-target samples) and specificity (describes the effectiveness of the classifier to classify correctly off-target samples). The classification was performed using the MATLAB machine learning package [56].

### 2.8. Prediction

The best model of each methodology was used to predict the Target % of the remaining spatial locations relative to each pass, as well as those used to build each method (Table 5). For this purpose, LFP signals related to each location were 5 s-segmented and features were extracted for each segment (Table 3), and then reduced using the number of principal components obtained for each methodology (13 for the Pass1 model and 11 for the Pass2 model). Our goal was to make use of the discriminative power of each classifier to distinguish between in- and off-target LFP signal physical locations, and consequently predict which electrodes were more inserted in the target, which may be the best choices for chronic stimulation.

## 3. Results

The Pass1 classifier (NN with 2 hidden layers) distinguished LFPs from DRight (in-target) and BLeft (off-target) spatial locations with an accuracy of 76.62%, sensitivity of 81.82% and specificity of 71.43%. The Pass2 classifier (kNN with a seuclidean distance) distinguished LFPs from ERight (in-target) and ALeft (off-target) physical locations with an accuracy of 83.33%, sensitivity of 100% and specificity of 66.6%. Predictions performed for each classifier are summarized in Table 6.

## 4. Discussion

Both classifiers (Pass1 and Pass2) classified the two class groups (in-target and out-target signals) with an average accuracy of 80%. Specificity, which quantifies the effectiveness of classifiers to identify correctly negative class labels (i.e., non-target signals), was also similar for both classifiers (70%, in average). Sensitivity, which classifies the effectiveness of classifiers to classify correctly positive class labels, i.e., target signals, was higher for the Pass2 classifier than Pass 1 (100% against 81.82%). Therefore, we considered that the Pass2 classifier had high performance, which was expected since spatial locations in Pass2 signals are closest to LFP channels.

The Pass1 classifier identified spatial locations completely outside the target (−) as 40% of the target; physical locations marginally in the target (+) as 50%; physical locations partially but not completely inserted in the target (++) as 50–70%. The spatial location completely inserted in the target (+++) was used as a classifier class label. Since the classifier results increased with the degree of involvement of each spatial location to the target, the Pass2 classifier predictions seem to agree with the medical imaging results (Figure 1).

The Pass2 classifier identified spatial locations completely outside the target (−) as 50% of the target; spatial locations marginally in the target (+) as 60%; spatial locations partially but not completely inserted in the target (++) as 50%. The spatial location completely inserted in the target (+++) was also used as a classifier label. These predictions seem to be worse than those obtained with the Pass1 classifier, since D_Left_ (+) was classified to be more target percentage than C_Right_ (++).

The target percentage of spatial locations used as class labels for each classifier were predicted by the opposite classifier (Table 5). Pass1 classified A_Left_ as 50% (which agrees with other predictions made by his classifier) and E_Right_ as 60% (higher target percentage obtained than other (++) spatial locations); The Pass2 classifier classified B_Left_ as 60% (higher target percentage than other (+) spatial locations) and D_Right_ as 60% (the same order of magnitude as other (++) predictions). Therefore, the Pass1 classifier seems to predict better Pass2 spatial locations than the opposite situation. This result was expected since spatial locations of nonadjacent channels include those for adjacent channels. Note that we considered Pass1 and Pass2 signals from different natures, since distance between channels and their midpoints vary according to the bipolar montage used. However, the only way to obtain a quantitative evaluation of the target prediction of spatial locations used to build each classifier is to use the opposite classifier. This is a limitation of the proposed methodology.

The gamma band power (30–100 Hz) was considered the most significant feature to discriminate between target and non-target signals in both montages and for the considered dataset (Figure 4). For the Pass2 classifier, the alpha band power (8–13 Hz) was equally relevant, whereas for the Pass1 classifier, all spectral features were considered discriminative. Considering morphological features, in the Pass2 classifier only the average mean had discriminative power; mean peaks and minimum values had middle discriminative power and the maximum value was the less discriminative. For the Pass1 classifier, morphological features were medially discriminative. Considering statistical features, variance and kurtosis seem to have good discriminative power for the Pass2 classifier, whereas for the Pass1 classifier only kurtosis appeared to have middle-range discriminative power. Considering multivariate features, the phase locking value had good discriminative power and cross-correlation was medially discriminative power in both classifiers. The phase lag index also discriminated well the two classes for the Pass2 classifier, and coherence had middle discriminative power. For the Pass1 classifier, phase lag index and coherence were not good features to discriminate between two classes.

## 5. Conclusions

In this work, we extracted 17 features from off-stimulation time-domain LFPs, recorded from one patient; and two ML-based methods, designed to identify which lead contacts presented a “signal signature” of the ANT target, the anteroventral (AV) nucleus. These methods were designed using different LFP bipolar montages: Pass1 (nonadjacent channels) and Pass2 (adjacent channels). After the best classification model for each dataset was identified, we obtained an accuracy of 76.62% for the Pass1 classifier and 83.33% for the Pass2 classifier. Then, we used both classifiers to predict the target percentage of all possible spatial locations, with the aim of identifying which contacts (in the left and right hemisphere) were closer or inserted to the target, and consequently, should be chosen for chronic stimulation.

We found that the spatial location E_Left_ had highest target percentage in the left hemisphere and spatial locations D_Right_ and E_Right_ in the right hemisphere. Converting spatial locations into channels (Table 2), we concluded that LFP signals from 2–3 L and 2–3 R and 1–3 R channels may present higher ANT target signatures. These results agree with those obtained by clinical images (Figure 1).

We concluded that the Pass1 and Pass2 classifiers were able to predict ANT target signatures for this patient. These results may be useful in support of the early selection of the best stimulation electrode, avoiding the trial-and-error process that usually takes place in the months after the implantation. To the best of our knowledge, we presented the first-ever study using multichannel LFPs collected months after the electrodes’ implantation (already out of the influence of the microlesion effect) in the guidance of the DBS target identification. However, further studies should be carried out in the future, with a larger number of patients, to validate this result.

## Figures and Tables

**Figure 1 sensors-22-06601-f001:**
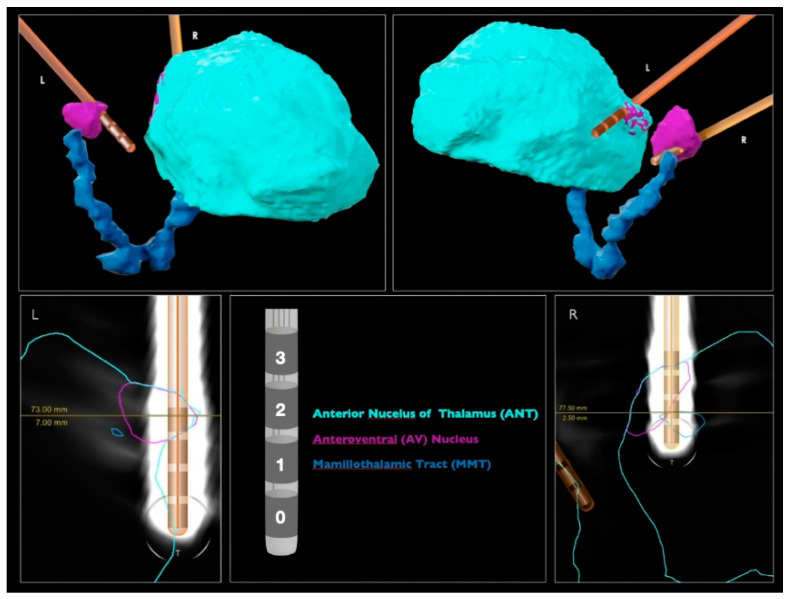
DBS lead localization performed by the BrainLab Elements. Images above show the perspective where the electrodes have a higher level of contact with the brain structures.

**Figure 2 sensors-22-06601-f002:**
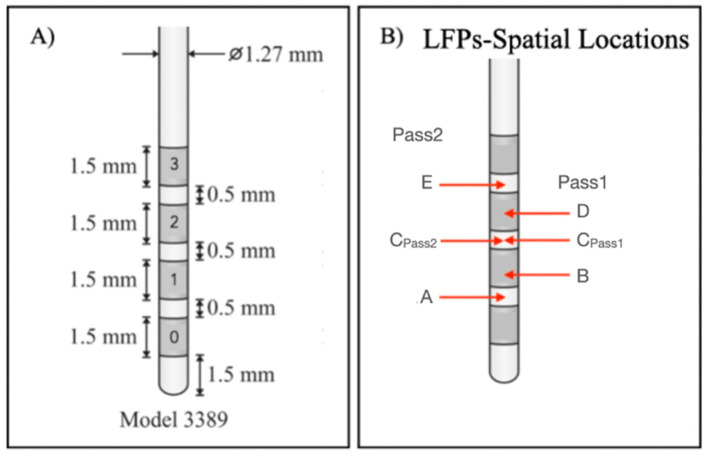
(**A**) Medtronic 3389 lead DBS model: each contact has a length of 1.5 mm, and the distance between two adjacent contacts is 0.5 mm; (**B**) LFP spatial locations of bipolar signals. Each spatial location corresponds to the midpoint between the considered channels.

**Figure 3 sensors-22-06601-f003:**
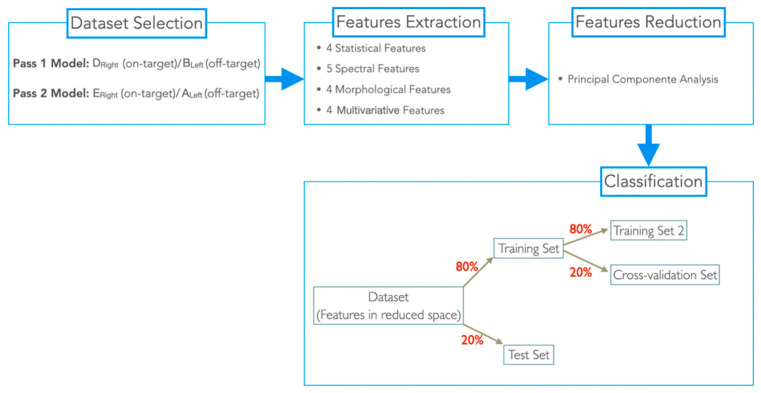
Schematic representation of the general methodology used.

**Figure 4 sensors-22-06601-f004:**
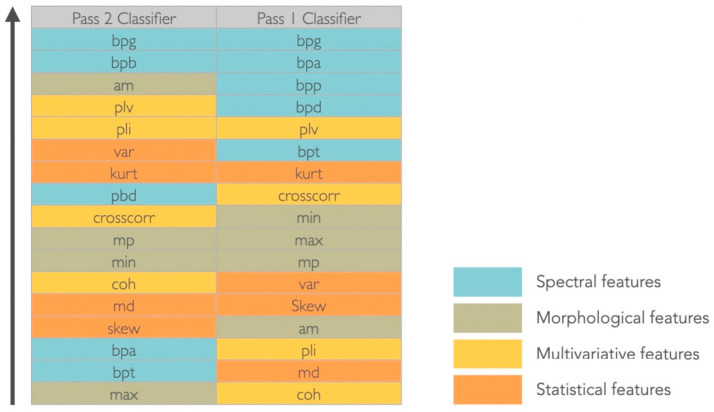
Ranking features results. Abbreviations—Spectral features: bpd (band power delta), bpt (band power theta), bpa (band power alpha), bpb (band power beta), bpg (band power gamma); Morphological features: am (absolute mean), mp (mean peaks), max (maximum), min (minimum); Multivariate features: crosscorr (cross-correlation), coh (coherence), plv (phase locking value), pli (phase lag index); Statistical features: var (variance), kurt (kurtosis), md (median), skew (skewness).

**Table 1 sensors-22-06601-t001:** Local field potential studies. Abbreviations: x—not applicable; STN—subthalamic nucleus; ACC—anterior cingulate cortex.

Reference	Description	Neurostimulator	Anatomical Target	Number of Patients
Jimenz-Shahed et al. (2021) [50]	Description of the sensing capabilities of the neurostimulator	Percept PC	x	x
Goyal et al. (2021) [49]	Description of the sensing capabilities of the neurostimulator	Percept PC	x	x
Yang et al. (2020) [48]	Study of which stimulation parameters provide pain relief without triggering after discharges	Activa PC+S	ACC	3
Passos et al. (2019) [47]	Study the impact of dopaminergic state and movement of beta functional connectivity between basal ganglia and lower motor neurons	Activa PC+S	STN	8
Anidi et al. (2019) [43]	Study whether STN-DBS affects beta burst dynamics and gait impairments differentially in freezers and non-freezers	Activa PC+S	STN	12
Hell et al. (2018) [44]	Study of the STN function during gait	Activa PC+S	STN	10
Maling et al. (2018) [45]	Development of a patient-specific computational framework to analyze LFP recordings	Activa PC+S	STN	1
Swann et al. (2018) [46]	Study of the disfunction of disturbed neural networks in FLPs	Activa PC+S	STN	5
Neumann et al. (2017) [41]	Study of long-term association of STN beta activity with parkinsonian motor signs	Activa PC+S	STN	15
Syrkin-Nikolau et al. (2017) [42]	Study of STN neural features of freezers and non-freezers	Activa PC+S	STN	14
Blumenfeld et al. (2016) [40]	Study whether the STN alpha/beta oscillation attenuation is causal to the bradykinesia improvement	Activa PC+S	STN	9
Quinn et al. (2015) [39]	Study whether beta power is similar in different resting postures during forward walking	Activa PC+S	STN	15

**Table 2 sensors-22-06601-t002:** Correspondence between bipolar LFP signal contacts, their spatial locations and the proportion of ANT target underlying each contact. Abbreviations—RH: right hemisphere; LH: left hemisphere; −: completely outside to the target; +: marginally in the target; ++: partially but not completely inserted in the target; +++: completely inserted in the target.

Contacts	Physical Location	Montage	Target % (LH)	Target % (RH)
0–1	A	Pass2	−	+
0–2	B	Pass1	−	+
1–2	C_Pass2_	Pass2	−	++
0–3	C_Pass1_	Pass1	−	++
1–3	D	Pass1	+	+++
2–3	E	Pass2	++	+++

**Table 3 sensors-22-06601-t003:** List of extracted features.

Feature Type	Feature	Description
Statistical	Variance (var)	Difference (spread) between the normalized squared sum of instantaneous values with the mean value
	Skewness (skew)	Distortion or asymmetry of the probability density function of the amplitude of time-series
	Kurtosis (kurt)	Sharpness of the probability density function of the amplitude of time-series
	Median (md)	Value separating the higher half from the lower half of a data sample
Spectral	Band power delta (bpd)	Power spectral density in the delta frequency band (1–3 Hz)
	Band power theta (pbt)	Power spectral density in the theta frequency band (4–7 Hz)
	Band power alpha (pba)	Power spectral density in the alpha frequency band (8–12 Hz)
	Band power beta (bpb)	Power spectral density in the beta frequency band (13–30 Hz)
	Band power gamma (bpg)	Power spectral density in the gamma frequency band (3–100 Hz)
Morphological	Absolute mean (am)	Absolute average value of a data sample
	Mean peaks (mp)	Absolute average of the maximum values of a data sample
	Maximum (max)	Maximum value of a data sample
	Minimum (min)	Minimum value of a data sample
Multivariate	Cross-correlation (crosscorr)	Measure of similarity of two time-series
	Coherence (coh)	Measures the causal relationship between two signals
	Phase locking value (plv)	Measure of the phase synchrony between two time-series
	Phase lag index (pli)	Evaluates the phase difference distribution across observations

**Table 4 sensors-22-06601-t004:** Classifiers and respective varying and standard model parameters.

Classifier	Model Parameters	Standard
SVM	Kernel function: linear, radial, basis function (rbf), polynomial	rbf
kNN	Distance metrics: squared Euclidean (seuclidean), Euclidean, correlation, Spearman	seuclidean
NN	Hidden layer size (1:10)	5

**Table 5 sensors-22-06601-t005:** Prediction of the target-% of lead DBS physical locations.

Model	Spatial Locations to Predict
Pass1	C_Pass1-Left_, C_Pass1-Right_, B_Right_, D_Left_, A_Left_ and E_Right_
Pass2	C_Pass2-Left_, C_Pass2-Right_, A_Right_, E_Left_, B_Left_ and D_Right_

**Table 6 sensors-22-06601-t006:** Pass1 and Pass2 classifiers ANT target percentage predictions. Pass1 classifier predicted ANT target percentages of Pass1 spatial locations (B_Right_, C_Pass1-Left_ and C_Pass1-Right_ and D_Left_) and Pass2 spatial locations used as a Pass2 classifier class labels (ALeft and ERight); Pass2 classifier predicted ANT target percentages of Pass2 spatial locations (A_Right_, C_Pass2-Left_ and C_Pass2-Right_ and E_Left_) and Pass1 spatial locations used as a Pass1-classifier class labels (B_Left_ and D_Right_). Gray cells indicate the predictions made by the opposite classifier. High prediction values for both hemispheres are written in bold. Abbreviations—LH: left hemisphere; RH: right hemisphere: completely outside the target; +: marginally in the target; ++: partially but not completely inserted in the target; +++: completely inserted in the target.

Spatial Location	Target (LH)	Predictions (LH)	Target (RH)	Predictions (RH)
A	−	54.35%	+	50.00%
B	−	63.04%	+	56.52%
C_Pass1_	−	41.30%	++	52.17%
C_Pass2_	-	52.17%	++	54.35%
D	+	40.87%	+++	**60.87%**
E	++	**69.57%**	+++	**60.87%**

## Data Availability

Not applicable.

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
