# Peer review of "Estimation of ANT-DBS Electrodes on Target Positioning Based on a New PerceptTM PC LFP Signal Analysis"

_sensors, 2022, doi:10.3390/s22176601_

Round 1

Reviewer 1 Report

This is an article with an interesting idea, however, with only one volunteer, which greatly limits its possibility of replication in patients. I suggest the authors increase the number of patients and send the manuscript again.

Author Response

  • The English has been revised and improved throughout the manuscript;
  • The introduction was divided into: (1) disease description; (2) ANT-DBS as an effective therapeutic approach; (3) Current methods used for target localization and its disadvantages; (4) scientific proposal: the use of LFPs in the guidance of DBS-targeting; (5) State of art - What is already known? ; (6) what we propose. A few clinical centers use acute, intraoperative microelectrode recordings (which record signal originating in single-cells, not LFPs) to assist electrode placement in the ANT. As can be concluded from Table 1, LFP studies are mainly focused on the study of biomarkers with the aim to develop closed-loop systems. Therefore, to the best of our knowledge, there is no sufficient background or relevant references supporting the study of LFPs in the guidance of DBS-targeting, which turns this preliminary study a pioneer in such area.
  • This study aims to open a new window on the study of LFPs (now that it is possible to record them chronically from such deep structures, such as the ANT) to assist in the DBS-target location. This, to our best knowledge, has never been researched and is of high importance since it is known that non-target structure stimulations lead to poor surgical outcomes. We developed a ML-based methodology to predict ANT-target signatures for one single patient and we obtained promising results. However, we know and have described in the manuscript, that this methodology must be reproduced and validated for a larger number of patients. Our objective is to share our methodology with the scientific community while new data is being collected in order to show our innovative and the capabilities of LFPs to assist in the DBS-target location. Even with only one patient, we believe that this study is scientifically pertinent, directing the scientific interest in the study of LFPs in the DBS-targeting.

Reviewer 2 Report

This paper examines whether ANT-LFPs recorded extra-operatively can 96 be an asset in the identification of DBS-target structure, by extracting 17 features using ML methods to identify lead contacts in both hemispheres relevant for ANT targets.

Overall, the authors posed an interesting and important question, and provided clear overview for motivation and related literature regarding DBS. However, this paper has a few limitations. 

1. While it's impressed to track LFP over 5 days, it's unclear whether signals remain robust and stable. How did the authors ensure signal quality and stability over 5 days? Even for longitudinal recording, the relative locations might shift and signal stability often degrades. 

2. The authors should provide a baseline for ML methods metrics in this study.

3. How many features were left after applying PCA? How do they map back to the original 17 features defined manually by the authors? 

Author Response

  • The English has been revised and improved throughout the manuscript;
  • The introduction was divided into: (1) disease description; (2) ANT-DBS as an effective therapeutic approach; (3) Current methods used for target localization and its disadvantages; (4) scientifically proposal: the use of LFPs in the guidance of DBS-targeting; (5) State of art - What is already known? ; (6) what we propose. As can be concluded from Table 1, LFP studies are mainly focused on the study of biomarkers with the aim to develop closed-loop systems. Therefore, to the best of our knowledge, there is no sufficient background or relevant references supporting the study of LFPs in the guidance of DBS-targeting;
  • The LFPs were recorded for a period of 5 days, 1 month after electrodes implementation, thus preventing the microlesion effect, as well as electrode displacements and brain shift immediately after implantation. The quality and stability of LFP signals, on the other hand, is ensured by existing technology (Percept PC neurostimulator). The signal is also recorded in a clinical controlled environment which allow us to have a higher degree of control of the patient movements.
  • The ML classifiers were developed for two class labels: signals coming from target vs signals coming from non-target structures (recorded under the same time and condition). These classifiers should be intra-patient. Since we did not find any related studies in the state of art (to our knowledge), it was not possible for us to make a comparison with other studies. Furthermore, our group has explored the classifiers used in this study for many different areas (e.g. Paiva et al., 2018 and Pereira et al., 2016).
  • With the PCA method, features were projected into another In this case, 17 features were projected into a 13- and 11-, we do not remap these new features into the original space.

Paiva, J. S., Cardoso, J., & Pereira, T. (2018). Supervised learning methods for pathological arterial pulse wave differentiation: a SVM and neural networks approach. International journal of medical informatics, 109, 30-38.

Pereira, Tânia, et al. "An automatic method for arterial pulse waveform recognition using KNN and SVM classifiers." Medical & biological engineering & computing 54.7 (2016): 1049-1059.

Reviewer 3 Report

Dear authors,

I appreciate the idea and technical realization presented in your study. LFPs have long been a very valuable source of information on neuronal populations' behaviors in multiple fields of neuroscience and they are well established as a fundamental aspect of brain research.

I find very interesting the combination of continuous „free” recording and the use of AI tools to extract those features that identify the targeted population of neurons. However, there are two aspects that I believe require further discussion and clarification in the conclusion.

First, as you mention this is a study on only one patient. Although acceptable to validate your method at the level of concept I believe further cases are needed to increase both the accuracy of the AI systems and the validity of the method as an established routine medical tool.

Second, although the thorough and obviously promising results, the method is quite a labor and expertise intensive to be easily adopted by other groups performing DBS. I believe that a description of an intended path to make an easily replicable and reliable method in the OR would be welcome. 

Author Response

  • The English has been revised and improved throughout the manuscript;
  • This study included one single patient, which is indeed a limitation. However, we aimed to demonstrate the potential use of LFPs in helping to identify the target of DBS (at the concept level). The methodology used is intra-patient. Nevertheless, we have been gathering efforts to replicate the methodology used for a larger number of patients. After this validation, the next step will be to automate the process and understand how this tool can be used to aid clinical decision. However, we agree that the limitation of having only one patient should be included in the conclusion. thus, we add the following sentence in the “Conclusions” section: “However, further studies should be carried out in the future, with a larger number of patients, in order to validate this result.”

Reviewer 4 Report

The idea presented by the authors is good since they present a technique that can help improve.

The reading of the summary is promising however, the article requires some changes

The presentation of keywords is a bit poor. Search engines use these terms for future searches and authors must improve the number of keywords.

The introduction is a bit complex. It presents parts that are part of the materials and methods and above all presents a lot of information that is relative to the discussion. Part of the introduction should go to the discussion and present a basic introduction.

They should rearrange the introduction and the part that is part of M&M and discussion put it in that part.

The results are clear and clearly indicate the proposed objectives.

The discussion is not really a discussion. Part of the introduction as I discussed earlier is really the discussion.

One change they must make is using the first person throughout the text. An impersonal use in the text is recommended.

Author Response

  • Thank you for the suggestion. It was poor so we added two more keywords: “Epilepsy” and “Closed-loop stimulation”
  • The introduction was divided into: (1) disease description; (2) ANT-DBS as an effective therapeutic approach; (3) Current methods used for target localization and its disadvantages; (4) scientifically proposal: the use of LFPs in the guidance of DBS-targeting; (5) State of art - What is already known?; (6) what we propose. We understand and appreciate your suggestion, however, we consider that the results of the state-of-the-art survey should remain in the introduction, as this is not the aim of the study, as well as it is our starting point for the exploratory study we did, as there are no similar studies in the literature
  • The discussion is based on the results achieved for the two classifiers. Also, we discussed which features were considered most relevant to distinguish LFPs patterns from non-target and target structures. Thus, we do not understand which part of the introduction should be replaced or added in the present discussion.
  • We removed the first person throughout the text.

Round 2

Reviewer 1 Report

After the revisions carried out by the authors, the article became clearer for the understanding of the readers. I believe it now presents conditions to be accepted for publication.

Author Response

Thanks.

Reviewer 4 Report

Thank you for the changes made to the manuscript.

The changes have improved the manuscript and made the proposed approaches more explicit.

Author Response

Thanks.